# Cross-sectional assessment of *Angiostrongylus cantonensis* transmission risk mediated by invasive apple snails in Jiangsu province of China

Qiang Zhang[1], Fanzhen Mao[1], Bixian Ni[1], Jiayao Zhang[1], Feng Tang[1], Xinyi Gong[1], Yaobao Liu[1,2]*

**1** National Health Commission Key Laboratory of Parasitic Disease Control and Prevention, Jiangsu Provincial Key Laboratory on Parasite and Vector Control Technology, Jiangsu Provincial Medical Key Laboratory, Jiangsu Institute of Parasitic Diseases Wuxi, Wuxi, Jiangsu, China, **2** School of Public Health, Nanjing Medical University, Nanjing, Jiangsu, China

* yaobao0721@163.com

## Abstract

### Background

*Angiostrongylus cantonensis*, a zoonotic nematode responsible for eosinophilic meningitis, is acquired in China principally through ingestion of raw or under-cooked invasive apple snails—the vector implicated in almost all Chinese outbreaks to date. Despite the widespread distribution of these snails in Jiangsu Province of China, systematic assessments of infection risks and public awareness remain limited.

### Methodology/Principle findings

We carried out a cross-sectional study in six prefectural-level cities in Jiangsu. In total, 732 wild apple snails were collected from different habitats and screened for *A. cantonensis* larvae by pepsin–HCl digestion and microscopy. 74 local farmers markets were inspected for the sale of live or processed apple snails. Simultaneously, a validated Knowledge, Attitudes, and Practices (KAP) questionnaire was conducted with 630 randomly selected residents. Data analysis employed SPSS for descriptive statistics and multivariate linear regression was used to identify predictors of higher scores. All wild-caught snails tested negative for *A. cantonensis*, and no apple snails were found for sale in any market. Overall, the public demonstrated a high level of knowledge and positive attitudes towards the risks posed by apple snails; however, gaps in practice were evident: 40.3% of respondents reported not separating utensils used for raw foods from those used for cooked foods, and 78.7% reported snail sightings to authorities, while only 22.1% had ever used the Apple Snail Inspector (ASI) mini-program. KAP scores differed significantly among different demographic

which permits unrestricted use, distribution, and reproduction in any medium, provided the original author and source are credited.

**Data availability statement:** All data analyzed during this study are included in this article.

**Funding:** This study was supported by the Jiangsu Province Capability Improvement Project through Science, Technology, and Education (No. ZDXYS202207) and Jiangsu Provincial Preventive Medicine Association Research Program (Y12023051). The funders had no role in the conceptualization, design, data collection, analysis, decision to publish, or preparation of the manuscript.

**Competing interests:** The authors have declared that no competing interests exists.

groups. Multivariable linear regression analysis revealed that age, city of residence, and education level were significant predictors of KAP scores.

## Conclusion/Significance

While no infected snails were detected in this study, the potential risk of transmission remains because of the widespread presence of competent apple snails. Targeted health education programs that address behavioral gaps among older and less-educated residents could further minimize the threat of *A. cantonensis* introduction and safeguard both public health and ecological security.

### Author summary

In this study, we investigated the risk of human infection with the rat lungworm, *Angiostrongylus cantonensis*, which can cause severe meningitis. This parasite is often spread through invasive apple snails, which are now widespread in Jiangsu Province, China. We combined field surveys of snails with a public questionnaire to understand both ecological and behavioral risks. Although we did not find any infected snails in the wild or in local markets, our survey revealed important gaps in public health practices. For example, many people do not separate utensils for raw and cooked food, and few use our reporting app for snail sightings. We also observed that older individuals and those with lower levels of education demonstrated lower awareness and exhibited less effective preventive habits. Our findings show that even without current detection of the parasite, the potential for outbreak remains because of widespread snails and unsafe behaviors. We recommend targeted health education, especially for high-risk groups, and continued monitoring to prevent future infections. This integrated approach will help protect both public health and local ecosystems.

## Introduction

*Angiostrongylus cantonensis*, the rat lungworm, first described by Chen in 1935 [1], is now recognized as the leading global cause of human eosinophilic meningitis [2]. Human infection occurs primarily through the intentional or accidental ingestion of third-stage larvae (L3) present in infected intermediate hosts or paratenic hosts. A potential, though less documented, route involves the ingestion of L3 larvae that may contaminate vegetables, possibly via contact with slugs or small snails, or, more theoretically, in water [3,4]. After ingestion, the L3 travel from the intestines to the brain, where upon death the L3 elicit a strong immune response, leading to meningitis and severe neurological symptoms, including headaches, cranial nerve palsies, and, in severe cases, coma or death [5–7]. Originally endemic to Southeast Asia and subsequently to Pacific Islands, the parasite has expanded its range to over 30 countries across Asia, Africa, the Americas, Europe, and Australia, driven by dispersal of rats

(definitive hosts) snails (intermediate hosts) associated with global commerce [8,9]. In recent years, reported neuroangio-strongyliasis cases have increased significantly in China [10–12]. Moreover, there have been multiple outbreaks, which pose a serious threat to human health and have attracted urgent public-health attention [13–16].

Human infections are driven largely by culinary habits, particularly the consumption of raw or under-cooked snail meat—fuelled by a growing appetite for exotic or "fresh" dishes. This behavior heightens exposure to L3, facilitating outbreaks of neuroangiostrongyliasis. In China, invasive apple snails (*Pomacea* spp.) play pivotal roles in transmitting *A. cantonensis* and almost all documented outbreaks have been traced to their ingestion [16–18]. Apple snails, now wide-spread across southern, southwestern, and central China, have been confirmed as a carrier in several provinces [19,20]. Jiangsu Province is abundant in water resources and the invasive apple snail has rapidly spread in this area, causing severe damage to aquatic ecosystems including farmlands and ponds in recent years [21]. It has been listed as one of the eight Level 1 controlled species in Jiangsu province's first catalog of invasive alien species [22]. To date, over 1,200 infestation sites have been documented by our real-time monitoring platform—Apple Snail Inspector (ASI) across various aquatic environments including farmlands, park wetlands, and riverine habitats [23]. Although no autochthonous human neuroangiostrongyliasis cases have yet been reported in Jiangsu, the confluence of widespread vector presence and risky food behaviors makes sustained monitoring and targeted health education an urgent public-health priority.

A recent survey has shown that approximately 27% of residents were unable to identify apple snails in Shanghai [24], and an online survey revealed that 10.6% of respondents would still knowingly eat raw food, despite the risk of parasitic infection [25]. These gaps between recognition and practice underscore the critical importance of foodborne disease prevention during the annual reproductive peak. The Knowledge, attitudes, and practices (KAP) model is a foundational tool in public health research. It offers a systematic approach to understanding health behaviors, empowering research-ers to design impactful interventions and assess their success. KAP surveys play a vital role in this process by assessing the current level of knowledge within a population regarding specific health issues [26]. By quantitatively analyzing public knowledge, preventive attitudes, and behavioral practices regarding apple snails, cognitive vulnerabilities can be precisely identified. Sustained KAP monitoring not only evaluates the effectiveness of awareness campaigns but also guides the establishment of a community-based "identification-prevention-reporting" through behavior change theories, ultimately forming a closed-loop control system [27].

Therefore, this study aimed to evaluate the infection status of *A. cantonensis* in invasive apple snails in Jiangsu Province, and simultaneously investigate the KAP of local residents regarding apple snails and *A. cantonensis* transmis-sion. Findings will be translated into precision interventions: risk-communication campaigns for low-awareness groups, enhanced alerts for those underestimating danger, and behavior-nudging measures to correct unsafe practices. These integrated efforts will help protect public health and, by enhancing community-based surveillance and control of the inva-sive apple snail, will also contribute to safeguarding local ecosystems.

## Methods

### Field sampling and detection of apple snails

Six prefectural-level cities in Jiangsu (Wuxi, Suzhou, Nantong, Yancheng, Taizhou, and Suqian) were selected to cover the province's major drainage basins and to coincide with ASI hot-spot data. ASI is a smartphone-based, crowd-sourced surveillance system developed by our institute for real-time reporting and monitoring of invasive apple snails [23]. The platform consists of a WeChat mini-program accessible to the public. Members of the public can report snail sightings by uploading geotagged photographs and information on the snail's habitat through the mini-program. These reports are instantly submitted to a centralized database managed by our research team. A minimum of four sampling sites per city were selected. However, in cities with fewer than four documented infestation sites, all available sites were enrolled, prioritizing locations with both apple snail infestations and signs of rodent activity. Snails were collected between June and September 2024 (the species' peak reproductive window) using a combination of dip-netting and gloved hand-picking. At

least 25 apple snails were collected per site, with a total sample size reaching ≥100 snails for subsequent detection. Collected snails were transported alive to the laboratory and identified morphologically as apple snails. To detect *A. cantonensis* infection, two standard parasitological techniques were employed: lung tissue microscopy and artificial digestion with pepsin-HCl [28]. Briefly, snails were euthanized by freezing at -20°C for 24 hours, followed by necropsy to microscopically examine lung tissue for larvae. The entire soft tissue of each snail was then digested in 1% pepsin in 0.7% HCl at 37°C for 2–3 hours with constant agitation. After digestion, the mixture was filtered through a 30-mesh sieve, and the sediment was examined under a microscope for L3 larvae of *A. cantonensis*.

During the monitoring period, field surveys were carried out at a minimum of three markets per site each month. If apple snails had been found on sale at these markets, specimens would have been purchased and subjected to laboratory detection. Unfortunately, none was found.

### Population-based KAP survey

We developed a 16-item KAP questionnaire tailored to the Jiangsu context (S1 Text). Questionnaire content validity was refined through a group meeting with experts in parasitology, epidemiology, health education, and veterinary public health. The questionnaire comprised three dimensions: (1) Knowledge: 5 items assessing recognition of apple snails, awareness of their health risks, and understanding of preventive measures (correct answer scored 1 and incorrect answer scored 0); (2) Attitude: 4 items evaluating risk perception and willingness to adopt protective behaviors (positive attitude scored 1 and negative attitude scored 0); (3) Practice: 7 items measuring self-reported behaviors related to snail handling, consumption, and reporting (good practice scored 1 and poor practice scored 0). The internal consistency reliability of the KAP questionnaire was assessed using Cronbach's alpha coefficient [29], a measure of how closely related a set of items are as a group (ranging from 0 to 1, with higher values indicating greater internal consistency). The tool demonstrated acceptable reliability (Cronbach's $\alpha = 0.63$) and was validated through pilot testing. The surveys were implemented in communities and schools adjacent to their infestation sites, with a requirement of surveying ≥ 100 respondents per monitoring site. Demographic variables included city of residence, age, gender, education, occupation, and marital status.

### Statistical analysis

Data were reported using a customized Epidata database [30] and entered using a dual-entry verification method by two independent operators. All visualizations were produced using Microsoft Excel 2016. Statistical analyses were performed with SPSS 23.0. Descriptive statistics (mean ± standard deviation, $\bar{x} \pm s$) summarized KAP scores across subgroups. Intergroup differences were evaluated via independent t-tests and one-way ANOVA. Multivariate linear regression identified predictors of KAP scores, adjusting for covariates such as city of residence, age, education, occupation and marital status. (A univariate $P < 0.05$ for the total score were included in the model). Statistical significance was set at $P < 0.05$ (two-tailed).

### Ethical considerations

This study was reviewed and approved by the Institutional Ethics Committee of Jiangsu Institute of Parasite Diseases. Written informed consent was obtained from all participants, and data were anonymized to ensure confidentiality.

## Results

### No infected apple snails and market sales observed

In total, 732 wild apple snails were collected across the six surveyed cities in Jiangsu Province (S1 Table): Wuxi (100), Suzhou (135); Nantong(112), Yancheng (160), Taizhou (100), Suqian (125). Laboratory detection revealed no *A. cantonensis* larvae in any specimen. Concurrently, market inspections across 74 local markets (Wuxi: 12; Suzhou: 12; Nantong:

12; Yancheng: 14; Taizhou: 12; Suqian: 12) found no apple snails offered for sale, indicating effective market regulation and a low risk of human exposure to infected snails in these regions.

**Responses of the surveyed population to the KAP questions regarding apple snails**

In total, 630 questionnaires were collected in this study. In the knowledge component, the percentage of correct answers was relatively high (>80% correct on all items), indicating that the public has a certain level of understanding in these areas. Regarding attitudes, most respondents showed a positive attitude with 98.3% of respondents expressing a desire to learn about apple snail hazards, and 97.3% willing to share relevant knowledge with others. In terms of practices, although a certain number of people separate cutting boards and knives for raw and cooked food, there is still a significant portion who do not (40.3%). Moreover, only 78.7% of respondents would report the discovery of apple snails to the authorities and just 22.1% had ever used the ASI platform (Fig 1).

**KAP scores across demographic subgroups**

KAP scores showed significant disparities among different demographic groups (Table 1). Univariate analysis revealed that scores varied significantly by city of residence, age, education level, occupation, and marital status. In contrast, no significant difference was found between genders. By city ($P<0.001$), Nantong exhibited the highest score ($14.10\pm1.92$), while Suqian had the lowest score ($12.44\pm2.20$). Age emerged as a critical factor: individuals aged $<20$ scored highest ($13.82\pm1.80$), while those $>40$ scored lowest ($12.93\pm2.08$) ($P<0.001$). Education level also significantly influenced scores, with higher education (e.g., college graduates: $13.91\pm1.84$) associated with better KAP scores compared to primary education ($13.11\pm2.09$) ($P<0.001$). Occupation-based analysis highlighted students ($13.83\pm1.77$) and "other" occupations ($13.84\pm1.49$) as high-performing groups, whereas farmers scored lowest ($12.59\pm2.12$) ($P<0.001$). Marital status further stratified results, with unmarried individuals ($13.75\pm1.90$) outperforming married participants ($13.12\pm2.04$) ($P<0.001$).

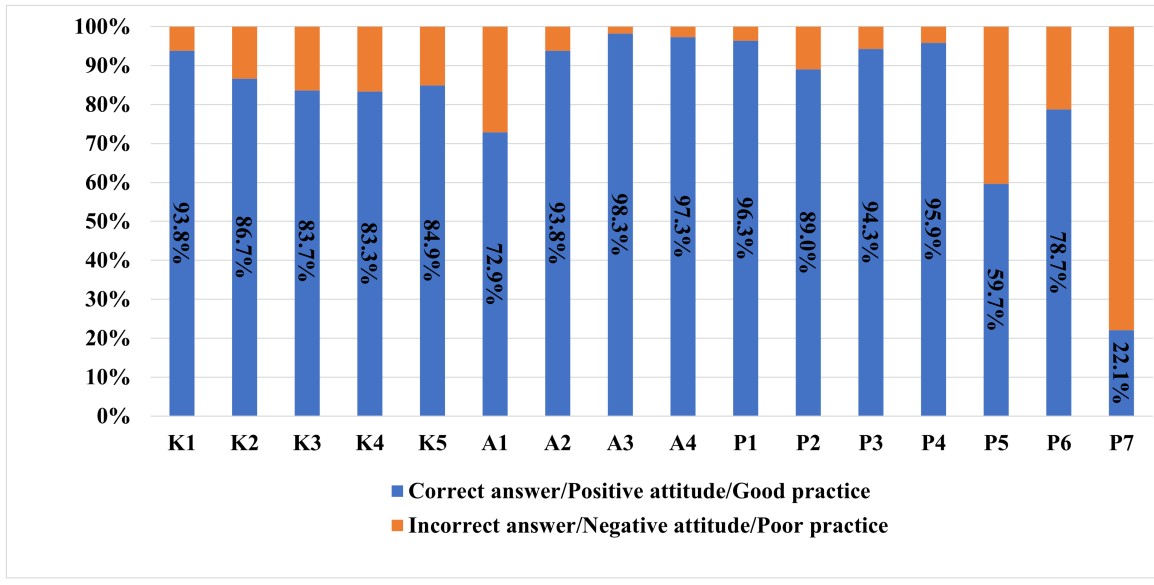

**Fig 1. Responses of the surveyed population to the knowledge, attitudes, and practices (KAP) questions about apple snails.**

**Table 1. Comparison of KAP scores in each dimension with different demographic characteristic.**

| Characteristics | Number of surveys | Knowledge ($\bar{x}\pm s$) | Attitude ($\bar{x}\pm s$) | Practice ($\bar{x}\pm s$) | Total ($\bar{x}\pm s$) |
|---|---|---|---|---|---|
| City | | | | | |
| Wuxi | 100 | 4.51±1.02 | 3.84±0.37 | 5.47±1.03 | 13.82±2.07 |
| Suzhou | 100 | 4.45±1.10 | 3.74±0.51 | 5.47±1.05 | 13.66±1.99 |
| Nantong | 105 | 4.43±1.28 | 3.88±0.36 | 5.80±0.58 | 14.10±1.92 |
| Yancheng | 116 | 4.94±0.30 | 3.87±0.45 | 4.86±1.05 | 13.67±1.25 |
| Taizhou | 100 | 3.38±1.18 | 3.24±0.57 | 5.50±1.16 | 12.12±1.81 |
| Suqian | 109 | 4.15±1.31 | 3.16±0.72 | 5.14±1.11 | 12.44±2.20 |
| F | | 24.488 | 44.494 | 11.471 | 19.330 |
| P | | <0.001* | <0.001* | <0.001* | <0.001* |
| Gender | | | | | |
| Male | 339 | 4.39±1.09 | 3.60±0.62 | 5.36±1.03 | 13.35±1.90 |
| Female | 291 | 4.24±1.27 | 3.65±0.57 | 5.36±1.09 | 13.25±2.15 |
| t | | 1.562 | -1.202 | 0.065 | 0.599 |
| P | | 0.119 | 0.230 | 0.949 | 0.549 |
| Age | | | | | |
| <20 | 152 | 4.77±0.90 | 3.73±0.59 | 5.32±0.95 | 13.82±1.80 |
| 20-40 | 135 | 4.47±1.02 | 3.61±0.53 | 5.60±0.96 | 13.68±1.92 |
| >40 | 343 | 4.07±1.27 | 3.58±0.62 | 5.28±1.12 | 12.93±2.08 |
| F | | 21.522 | 3.440 | 4.565 | 13.793 |
| P | | <0.001* | 0.033* | 0.011* | <0.001* |
| Education level | | | | | |
| Primary school and below | 230 | 4.27±1.29 | 3.61±0.65 | 5.22±1.05 | 13.11±2.09 |
| Junior high school | 150 | 4.09±1.29 | 3.55±0.62 | 5.10±1.10 | 12.74±2.09 |
| High school or technical secondary school | 99 | 4.42±0.91 | 3.67±0.54 | 5.61±0.99 | 13.70±1.68 |
| Junior college and above | 151 | 4.56±0.97 | 3.68±0.51 | 5.67±0.95 | 13.91±1.84 |
| F | | 4.452 | 1.527 | 10.975 | 10.960 |
| P | | 0.004* | 0.206 | <0.001* | <0.001* |
| Occupation | | | | | |
| Farmer | 149 | 3.95±1.37 | 3.55±0.63 | 5.09±1.19 | 12.59±2.12 |
| Worker | 117 | 4.20±1.18 | 3.63±0.60 | 5.32±0.99 | 13.15±1.96 |
| Medical Worker | 58 | 4.34±1.15 | 3.57±0.62 | 5.55±1.13 | 13.47±2.20 |
| Self-employed | 64 | 4.22±1.24 | 3.59±0.61 | 5.33±1.156 | 13.14±2.38 |
| Student | 157 | 4.78±0.89 | 3.71±0.59 | 5.34±0.94 | 13.83±1.77 |
| Others | 85 | 4.39±0.95 | 3.62±0.49 | 5.82±0.76 | 13.84±1.49 |
| F | | 8.663 | 1.308 | 5.850 | 7.706±2.02 |
| P | | <0.001* | 0.259 | <0.001* | <0.001* |
| Marital Status | | | | | |
| Unmarried | 186 | 4.69±1.00 | 3.70±.59 | 5.36±0.97 | 13.75±1.90 |
| Married | 444 | 4.17±1.21 | 3.59±.59 | 5.36±1.09 | 13.12±2.04 |
| t | | 5.542 | 2.253 | -0.002 | 3.626 |
| P | | <0.001* | 0.025* | 0.999 | <0.001* |

* Statistically significant.

## Multivariate predictors of KAP scores

Variables with a univariate *p*-value of total scores below 0.05 were further analyzed using multivariate linear regression analysis to identify key predictors of KAP scores. Three factors remained significant predictors of total KAP score: city of residence (all cities scored higher than the Suqian reference group. Suqian was chosen as the reference category for the city variable as it demonstrated the lowest mean KAP scores), younger age and higher education level (Table 2), whereas occupation and marital status did not independently predict scores in the adjusted model.

## Discussion

The present findings provide a critical assessment of the current status of *A. cantonensis* infection risk mediated by invasive apple snails in Jiangsu Province of China, and integrates snail surveillance with a detailed analysis of human behavior (KAP). Although no infected snails were detected, the potential for transmission persists due to widespread vectors and unsafe practices, underscoring that current absence should not be equated with enduring safety. The integration of field data and behavioral insights provides policymakers with an evidence-based dual perspective on both ecological and behavioral dimensions to design targeted interventions and strengthen health education.

Although invasive apple snails are a significant vector for *A. cantonensis* [31], and are widely distributed throughout Jiangsu's water networks [23,32], our surveillance did not detect the parasite in any collected specimens. Multiple provinces adjacent to Jiangsu have documented infections of *A. cantonensis* in apple snails [33,34]. The apparent absence in Jiangsu may be attributable to several factors. Although our sample size provides a preliminary overview, it may be insufficient to detect a low prevalence of infection. Furthermore, our sampling was conducted during a single season (the summer reproductive peak), which might not capture the full dynamic of the parasite's life cycle. Lastly, presently unfavorable

**Table 2. Multivariate linear regression analysis of the influence of different characteristics on KAP scores.**

| Characteristics | β | Standard error | Standardized β | P |
|---|---|---|---|---|
| Age | -0.030 | 0.010 | -0.307 | 0.002* |
| Marital Status (Ref = Married) | -0.461 | 0.372 | -0.104 | 0.216 |
| City (Ref = Suqian) | | | | |
| Wuxi | 1.214 | 0.275 | 0.220 | <0.001* |
| Suzhou | 1.265 | 0.267 | 0.229 | <0.001* |
| Nantong | 2.025 | 0.259 | 0.374 | <0.001* |
| Yancheng | 1.683 | 0.263 | 0.323 | <0.001* |
| Taizhou | 0.609 | 0.286 | 0.110 | 0.034 |
| Education level (Ref = Primary School and Below) | | | | |
| Junior high school | 0.484 | 0.226 | 0.102 | 0.033* |
| High school or technical secondary school | 1.342 | 0.288 | 0.242 | <0.001* |
| College and above | 1.274 | 0.333 | 0.270 | <0.001* |
| Occupation (Ref = Farmer) | | | | |
| Worker | -0.494 | 0.292 | -0.095 | 0.091 |
| Medical Worker | -0.702 | 0.377 | -0.101 | 0.063 |
| Self-employed | -0.329 | 0.331 | -0.049 | 0.322 |
| Student | 0.057 | 0.538 | 0.012 | 0.915 |
| Others | 0.071 | 0.328 | 0.012 | 0.829 |

*Statistically significant.

local ecological conditions, such as low densities of definitive rodent hosts or climatic factors suboptimal for larval development in snails, could also contribute to the absence of detected infection [35]. The absence of apple snails in the 74 surveyed markets suggests effective local regulation, which likely reduces the direct risk of human infection through commercial consumption. However, the morphological overlap between apple snails and *Cipangopaludina chinensis* [36] raises the possibility of undetected misidentification during the market surveys. Therefore, continuous surveillance, as recommended by the World Health Organization for neglected tropical diseases, is essential to detect early-stage zoonotic shifts [37].

The KAP survey results show that the residents in Jiangsu have a relatively high knowledge level of apple snails. This could be attributed to previous health education efforts and increased public awareness of food-borne diseases [26,38]. However, a small proportion of the population remains unaware of apple snails and their associated risk. In terms of attitude, most respondents had a positive attitude, such as a strong willingness to learn about apple snail hazards and disseminate health information, providing a receptive basis for future interventions. However, previous research has shown that knowledge does not reliably translate into preventive behaviors, especially when there is a lack of perceived personal risk or motivation [39]. Although a certain number of households separated raw and cooked cutting boards and knives, 40.3% did not. While the direct risk of *A. cantonensis* transmission via cross-contamination from utensils is likely lower than via direct ingestion of infected hosts, this practice gap reflects a broader potential for general foodborne pathogen transmission. It also highlights a crucial area for health education, as improving fundamental food safety practices can concurrently mitigate risks associated with various parasitic and bacterial diseases [40]. Moreover, the low reporting rates of apple snail discoveries (78.7% to authorities and 22.1% using ASI) underscore the need for improvement in public participation in surveillance.

Demographic disparities in KAP scores were pronounced in this study. Respondents aged < 20, those with a higher level of education, students, and unmarried individuals generally had higher scores, perhaps because younger age groups access diverse information channels, and higher education fosters stronger information-seeking and risk-perception skills. In contrast, farmers had the lowest scores, which could be related to their limited access to health education resources and different living and working environments. This is consistent with previous research indicating that education level is a robust determinant of health knowledge and behavior [26,41]. These results highlight the need for a differentiated health education strategy. For example, concise, actionable training modules and pictorial materials should be designed specifically for farmers, delivered through village health workers to maximize reach and impact.

Multivariate regression analysis identified age, city of residence, and education level as the significant predictors of KAP scores. Furthermore, the significant differences in KAP scores across cities highlight the need for region-specific strategies. Cities with higher KAP scores such as Nantong, may benefit from reinforcing existing educational programs, while cities with lower scores like Suqian, require more intensive interventions. This approach aligns with the principles of community-based participatory research [42], which emphasizes the importance of local context in designing effective public health initiatives. The current lack of specific public health guidelines for *A. cantonensis* in many at-risk regions, including newly endemic areas [43], further highlights the urgency of developing such tailored, evidence-based interventions to translate knowledge into sustained preventive behaviors. Additionally, promoting education-based prevention measures, such as incorporating apple snail related knowledge into school curricula and community education programs [44,45], could offer a scalable, education-centered pathway to raise public awareness and convert knowledge into sustained preventive behaviors.

This study has several limitations. First, the cross-sectional design precludes causal inferences between KAP factors and infection risks. Second the sample size is insufficient and the sampling focused on urban and peri-urban areas; rural regions with higher agricultural activity may face distinct challenges. Third, because all specimens were screened solely by pepsin–HCl digestion and microscopy, infections with very low larval burdens may have been missed [46]. Finally, it is important to emphasize that this survey targeted only the invasive apple snail, a recognized intermediate host; failure to detect *A. cantonensis* in this species does not imply its absence from other local intermediate or paratenic hosts.

Therefore, longitudinal studies are essential to monitor behavioral changes following interventions, and future research must extend to broader geographical regions (especially rural and remote areas) to enable a comprehensive assessment of both ecological and behavioral risks. Meanwhile, deploying molecular tools such as conventional PCR, qPCR or LAMP in future surveys will improve detection sensitivity and allow more precise monitoring of the parasite's transmission dynamics. Additionally, future surveillance should be broadened to include a wider range of potential hosts to provide a more comprehensive assessment of the local transmission risk and to understand the role of different species in sustaining the parasite's life cycle.

## Conclusion

This study highlights the importance of public awareness and behavior in preventing *A. cantonensis* infection. Although the parasite was not detected in Jiangsu Province in this survey, the persistent gaps in public knowledge and practices underscore the need for sustained efforts to improve food safety and disease prevention. A sustained, integrated strategy combining real-time surveillance, targeted education tailored to high-risk population, and evidence-based policy measures could safeguard public health in an era of increasing zoonotic threats.

## Supporting information

**S1 Text. Questionnaire on knowledge, attitudes, and practices (KAP) regarding apple snails in Jiangsu province.**
(DOCX)

**S1 Table. Investigation on the field infection and market sales status of apple snails.**
(DOCX)

## Author contributions

**Conceptualization:** Qiang Zhang, Fanzhen Mao, Bixian Ni, Jiayao Zhang, Feng Tang, Xinyi Gong, Yaobao Liu.

**Data curation:** Qiang Zhang, Jiayao Zhang.

**Formal analysis:** Qiang Zhang.

**Funding acquisition:** Qiang Zhang, Yaobao Liu.

**Investigation:** Qiang Zhang, Fanzhen Mao, Bixian Ni, Jiayao Zhang, Feng Tang, Xinyi Gong, Yaobao Liu.

**Methodology:** Qiang Zhang, Yaobao Liu.

**Project administration:** Qiang Zhang, Fanzhen Mao, Bixian Ni, Jiayao Zhang, Feng Tang, Xinyi Gong, Yaobao Liu.

**Resources:** Qiang Zhang, Fanzhen Mao, Bixian Ni, Jiayao Zhang, Feng Tang, Xinyi Gong, Yaobao Liu.

**Software:** Yaobao Liu.

**Supervision:** Qiang Zhang, Yaobao Liu.

**Validation:** Qiang Zhang, Yaobao Liu.

**Visualization:** Qiang Zhang, Yaobao Liu.

**Writing – original draft:** Qiang Zhang.

**Writing – review & editing:** Yaobao Liu.

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
