## [Decision Letter · Decision Letter 0]

12 Nov 2025

Cross-sectional assessment of Angiostrongylus cantonensis transmission risk mediated by invasive apple snails in Jiangsu Province of China

Dear Dr. LIU,

Thank you for submitting your manuscript to PLOS Neglected Tropical Diseases. After careful consideration, we feel that it has merit but does not fully meet PLOS Neglected Tropical Diseases's publication criteria as it currently stands. Therefore, we invite you to submit a revised version of the manuscript that addresses the points raised during the review process.

Please submit your revised manuscript within by DECEMBER 1ST. If you will need more time than this to complete your revisions, please reply to this message or contact the journal office at plosntds@plos.org. Please include the following items when submitting your revised manuscript:

We look forward to receiving your revised manuscript.

Kind regards,

Angela Monica Ionica, Ph.D.

Academic Editor

Krystyna Cwiklinski

Section Editor

Shaden Kamhawi

co-Editor-in-Chief

Paul Brindley

co-Editor-in-Chief

**Journal Requirements:**

At this stage, the following Authors/Authors require contributions: Qiang Zhang, Fanzhen Mao, Bixian Ni, Jiayao Zhang, Feng Tang, Xinyi Gong, and Yaobao Liu. Please ensure that the full contributions of each author are acknowledged in the "Add/Edit/Remove Authors" section of our submission form.

2) We notice that your supplementary Table is included in the manuscript file. Please remove it from the main file of the manuscript as it should be uploaded separately with the file type 'Supporting Information'. 

3) Please amend your detailed Financial Disclosure statement. This is published with the article. It must therefore be completed in full sentences and contain the exact wording you wish to be published.

2) State what role the funders took in the study. If the funders had no role in your study, please state: "The funders had no role in study design, data collection and analysis, decision to publish, or preparation of the manuscript.".

**Reviewers' Comments:**

**Comments to the Authors:**

**Please note that two reviews are uploaded as attachments.**

Reviewer's Responses to Questions

**Key Review Criteria Required for Acceptance?**

**Methods**

-Are the objectives of the study clearly articulated with a clear testable hypothesis stated?

-Is the study design appropriate to address the stated objectives?

-Is the population clearly described and appropriate for the hypothesis being tested?

-Is the sample size sufficient to ensure adequate power to address the hypothesis being tested?

-Were correct statistical analysis used to support conclusions?

-Are there concerns about ethical or regulatory requirements being met?

Reviewer #1: (No Response)

Reviewer #2: Yes

Reviewer #3: -Are the objectives of the study clearly articulated with a clear testable hypothesis stated? YES

-Is the study design appropriate to address the stated objectives? YES

-Is the population clearly described and appropriate for the hypothesis being tested? YES

-Is the sample size sufficient to ensure adequate power to address the hypothesis being tested? PERHAPS

-Were correct statistical analysis used to support conclusions? I HAVE CONCERNS ABOUT THE MULTIVARIATE ANALYSIS - NEED MORE DETAILS

-Are there concerns about ethical or regulatory requirements being met? NO

**Results**

-Does the analysis presented match the analysis plan?

-Are the results clearly and completely presented?

-Are the figures (Tables, Images) of sufficient quality for clarity?

Reviewer #1: (No Response)

Reviewer #2: Yes

Reviewer #3: -Does the analysis presented match the analysis plan? YES

-Are the results clearly and completely presented? SOMEWHAT - SEE MY COMMENTS ON THE ATTACHED MANUSCRIPT

-Are the figures (Tables, Images) of sufficient quality for clarity? TEXT ON THE FIGURE COULD BE LARGER FOR READABILITY

**Conclusions**

-Are the conclusions supported by the data presented?

-Are the limitations of analysis clearly described?

-Do the authors discuss how these data can be helpful to advance our understanding of the topic under study?

-Is public health relevance addressed?

Reviewer #1: (No Response)

Reviewer #2: Yes

Reviewer #3: -Are the conclusions supported by the data presented? UP TO A POINT - SEE MY COMMENTS ON THE ATTACHED MANUSCRIPT

-Are the limitations of analysis clearly described? YES - TO THE EXTENT THAT THE PAPER MAY NOT BE ACCEPTABLE

-Do the authors discuss how these data can be helpful to advance our understanding of the topic under study? YES

-Is public health relevance addressed? YES

**Editorial and Data Presentation Modifications?**

Reviewer #1: (No Response)

Reviewer #2: (No Response)

Reviewer #3: Most of my comments are of minor detail, including minor adjustments to the English. All are on the marked-up Word file, which I attach and that should be sent to the authors (the Word file not a pdf as conversion from Word to pdf will truncate at least some of the comments).

**Summary and General Comments**

Reviewer #1: Although this is an interesting study, in my opinion, the results are not scientifically robust enough to warrant publication in this journal. The manuscript does not provide substantial new insights into the ecological or health-related drivers of Angiostrongylus cantonensis transmission. Therefore, I would suggest that the authors consider submitting the article to PLOS Global Public Health, where it may be a better fit in terms of scope and focus. Please find my comments in the attached file.

Reviewer #2: 1. KAP should be defined the first time it is used, in the abstract

2. Why is T1 considered as Supplemental? It should be just Table 1 in the paper and adjust other accordingly.

3. L201, big space unneeded

4. L199, Why is 78.7% respondents referred to as ‘only’ 78.7%. That is a pretty big number of respondents

5. L 238 Indent needed for consistency

6. L 272.Again, why is 78.7% considered as a ‘low’ reporting rate, to me seems pretty high.

7. L290 Remove either ‘such as’ or ‘like’, so don’t need both.

8. Why wasn’t PCR testing conducted, it’s much more sensitive?

Reviewer #3: This is an unusual paper. It's in two parts: 1) a survey of snails in the wild and a search for snails for sale in markets to detect the rat lungworm parasite (Angiostrongylus cantonensis), which uses snails as intermediate hosts and is the cause of eosinophilic meningitis in humans, and 2) a KAP survey of people's responses to various questionsa about apple snails.

Regarding (1) no wild snails were found to be infected and no snails were even seen in the markets, so one could argue that there is no risk - the paper reports a non-result. Despite this, regarding (2) the KAP survey showed that people were aware to some extent of the possibility of snails being infected and that they could transmit rat lungworm, so if either infected snails were to spread into the region or the resident snails should become infected by transmission from incoming infected rats, the people are to an extent aware of the risks, receptive to education, and prepared to take appropriate measures to reduce the risk. Thus the KAP survey had some public health value. However, the major limitation of the wild snail survey was that only visual inspection of dissected snails was used to detect infection. A molecular genetic screening method would be much more sensitive and it is possible that it may have detected infected snails.

PLOS authors have the option to publish the peer review history of their article (what does this mean? ). If published, this will include your full peer review and any attached files.

**Do you want your identity to be public for this peer review?** For information about this choice, including consent withdrawal, please see our Privacy Policy .

Reviewer #1: No

Reviewer #2: No

Reviewer #3: No

**Figure resubmission:**
---

## [Editor Report · Decision Letter 1]

27 Nov 2025

Dear DR LIU,

We are pleased to inform you that your manuscript 'Cross-sectional assessment of Angiostrongylus cantonensis transmission risk mediated by invasive apple snails in Jiangsu Province of China' has been provisionally accepted for publication in PLOS Neglected Tropical Diseases.

Best regards,

Angela Monica Ionica, Ph.D.

Academic Editor

Krystyna Cwiklinski

Section Editor

Shaden Kamhawi

co-Editor-in-Chief

Paul Brindley

co-Editor-in-Chief

---

## [Editor Report · Acceptance letter]

Dear Mr. Liu,

We are delighted to inform you that your manuscript, "Cross-sectional assessment of Angiostrongylus cantonensis transmission risk mediated by invasive apple snails in Jiangsu Province of China," has been formally accepted for publication in PLOS Neglected Tropical Diseases.

Best regards,

Shaden Kamhawi

co-Editor-in-Chief

Paul Brindley

co-Editor-in-Chief
